# Sequential Blockade of PD-1 and PD-L1 Causes Fulminant Cardiotoxicity—From Case Report to Mouse Model Validation

**DOI:** 10.3390/cancers11040580

**Published:** 2019-04-24

**Authors:** Shin-Yi Liu, Wen-Chien Huang, Hung-I Yeh, Chun-Chuan Ko, Hui-Ru Shieh, Chung-Lieh Hung, Tung-Ying Chen, Yu-Jen Chen

**Affiliations:** 1Department of Medical Research, MacKay Memorial Hospital, New Taipei City 25160, Taiwan; syliu0830@gmail.com (S.-Y.L.); ru123@mmh.org.tw (H.-R.S.); 2Department of Thoracic Surgery, MacKay Memorial Hospital, Taipei 10449, Taiwan; wjhuang0@yahoo.com.tw; 3Department of Internal Medicine, MacKay Memorial Hospital, Taipei 10449, Taiwan; yehmmc@mmc.edu.tw (H.-I.Y.); kochunchuan1113@gmail.com (C.-C.K.); jotaro3791@gmail.com (C.-L.H.); 4Department of Pathology, MacKay Memorial Hospital, Taipei 10449, Taiwan; wax921@gmail.com; 5Department of Radiation Oncology, MacKay Memorial Hospital, Taipei 10449, Taiwan; 6Department of Chinese Medicine, China Medical University Hospital, Taichung 40402, Taiwan

**Keywords:** check point inhibitors, programmed cell death protein 1, programmed cell death 1 ligand 1, cardiotoxicity, lung metastasis

## Abstract

The combined administration of programmed cell death 1 (PD-1) and programmed cell death ligand 1 (PD-L1) inhibitors might be considered as a treatment for poorly responsive cancer. We report a patient with brain metastatic lung adenocarcinoma in whom fatal myocarditis developed after sequential use of PD-1 and PD-L1 inhibitors. This finding was validated in syngeneic tumor-bearing mice. The mice bearing lung metastases of CT26 colon cancer cells treated with PD-1 and/or PD-L1 inhibitors showed that the combination of anti-PD-1 and anti-PD-L1, either sequentially or simultaneously administered, caused myocarditis lesions with myocyte injury and patchy mononuclear infiltrates in the myocardium. A significant increase of infiltrating neutrophils in myocytes was noted only in mice with sequential blockade, implying a role for the pathogenesis of myocarditis. Among circulating leukocytes, concurrent and subsequent treatment of PD-1 and PD-L1 inhibitors led to sustained suppression of neutrophils. Among tumor-infiltrating leukocytes, combinatorial blockade increased CD8^+^ T cells and NKG2D^+^ T cells, and reduced tumor-associated macrophages, neutrophils, and natural killer (NK) cells in the lung metastatic microenvironment. The combinatorial treatments exhibited better control and anti-PD-L1 followed by anti-PD-1 was the most effective. In conclusion, the combinatory use of PD-1 and PD-L1 blockade, either sequentially or concurrently, may cause fulminant cardiotoxicity, although it gives better tumor control, and such usage should be cautionary.

## 1. Introduction

The efficacy of immunotherapies that use antibodies to block programmed cell death 1 (PD-1) or its ligand 1 (PD-L1) have been extensively investigated for a variety of cancer types [1]. These types of immunotherapy, immune checkpoint inhibitors (ICIs), have been proved effective in advanced/metastatic non-small cell lung cancer (NSCLC) [2], colorectal cancer with high mismatch repair deficiency or microsatellite instability [3], and others [4,5]. In clinical practice, sequential shifting from a PD-1 inhibitor to its ligand, a PD-L1 inhibitor, or re-treatment with immunotherapy due to ineffectiveness or toxicity, is becoming more common to prolong survival in terminally ill patients [6].

The benefits of ICIs can be offset by the serious immune-related adverse events (irAEs), which mostly involve damage to the dermatologic, gastrointestinal, endocrine, respiratory, hepatic, and musculoskeletal systems [7,8]. So far, irAEs are reported and often encountered in patients treated with ipilimumab, an anti-CTLA-4 (cytotoxic T-lymphocyte–associated antigen 4) therapy [9,10], and are less frequent in patients treated with anti-PD-1 agents [11,12] or anti-PD-L1 antibodies [13,14]. A combination of ICIs has demonstrated a significant benefit to overall survival compared to monotherapy [15,16]. However, unexpected toxicities mediated by irAEs are also significantly higher using a combined ICI strategy [17,18], and require early detection and appropriate management [19]. Cardiotoxicity is regarded as a rare event of irAEs after ICI treatment [20,21,22]; nevertheless, recent reports indicate that immune-related myocarditis might be a serious and underestimated complication of immunotherapy [23,24,25]. Inflammation-mediated cardiotoxic effects can include myocarditis, pericarditis, perimyocarditis, left ventricular dysfunction without myocarditis, Takotsubo syndrome, and others [26]. The diagnosis of immune-related myocarditis should exclude viral and autoimmune myocarditis, coronary artery disease, and pulmonary embolism. Cardiac magnetic resonance imaging (MRI) to detect myocardial edema and late gadolinium enhancement is important, but sensitivity is currently inadequate [27,28]. Endomyocardial biopsy could be considered for tissue proof in clinical practice.

Here, we report a patient in whom fatal myocarditis developed after sequential use of PD-1 and PD-L1 inhibitors. To validate this finding, an immune competent tumor-bearing mouse model was used for the evaluation of myocarditis and immune responses.

## 2. Experimental Section

### 2.1. Patient Record

Study of the patient’s medical records was approved by the Institute Review Board of Mackay Memorial Hospital, Taipei, Taiwan. Treatment courses and clinical features of this patient were collected. 

### 2.2. Cell Culture

The mouse colorectal adenocarcinoma cell line CT26 was purchased from the American Type Culture Collection (Manassas, VA, USA) and maintained in Roswell Park Memorial Institute (RPMI)-1640 medium supplemented with 10% heat-inactivated fetal calf serum (FCS, Hyclone, Logan, UT, USA) and l-glutamine (200 mM). Stable clones expressing luciferase (CT26-Luc cells) were established by transducing lentivirus containing the North American Firefly Luciferase gene under the control of the SV40 promoter. CT26-Luc cells were maintained in the above medium containing G418 (500 μg/mL; Sigma-Aldrich, St. Louis, MO, USA).

### 2.3. Experimental Animal Model for ICI Treatment

Four-week-old male BALB/c mice were purchased from the Animal Resource Center of the National Science Council of Taiwan (Taipei, Taiwan). All animal experiments were approved by the Animal Ethics Committee of Mackay Memorial Hospital (Taipei, Taiwan). Approval was received on 24 January 2019 and the IRB approval number is 19MMHIS008e. CT26-Luc cells (1 × 10^6^) were injected into tail veins of Balb/c mice to establish lung metastasis. The IVIS 200 imaging system (Xenogen Biosciences, Cranbury, NJ, USA) was used to estimate lung metastatic burden. After lung metastasis was established, the anti-mouse PD-1 monoclonal antibody (200 μg per mouse, RMP1-14, BioXcell, Lebanon, NH, USA), anti-mouse PD-L1 monoclonal antibody (150 μg per mouse, 10F.9G2, BioXcell, Lebanon, NH, USA), or their isotype control mAbs (rat IgG2a or rat IgG2b) were intraperitoneally administered. For concurrent combination, the anti-PD-1 and anti-PD-L1 mAbs were injected on the same days (days 0, 2, and 4). For subsequent combination, the second mAb was administered on days 6, 8, and 10. Mice were sacrificed on day 11 or day 17 for tumor microenvironment assessment as described in the text.

### 2.4. Immunohistochemistry (IHC) and Grading of Myocarditis

Sections of formalin-fixed and paraffin-embedded mouse hearts were treated with heat-induced antigen retrieval in sodium citrate buffer (10 mM sodium citrate, 0.05% Tween 20, pH 6.0) at 95−100 °C for 15 min, followed with 3% H_2_O_2_ at room temperature for 10 min. Blocking was performed with BlockPROTM Protein Blocking Buffer (Visual Protein Biotech., Taipei, Taiwan) at room temperature for 1 h. Sections were incubated with primary antibodies against Ly6C (1:200; Abcam, Cambridge, UK) and Ly6G (1:100; Abcam, Cambridge, UK) at room temperature for 1 h, with secondary antibodies of SignalStain^®^ Boost IHC Detection Reagent and Rabbit Anti-Rat IgG (1:200; Abcam, Cambridge, UK) at room temperature for 1 h, and developed for 5−10 min with DAB (3,3’-Diaminobenzidine) chromogen kit (EnVision™+ Dual Link System-HRP, DAKO, Carpinteria, CA, USA) before counterstaining with hematoxylin (Merck, Darmstadt, Germany). The proportion of cells with Ly6C and Ly6G staining were calculated from a high-power field for 10 different portions by microscopy. Sections with 4 μm thickness were stained with hematoxylin and eosin (H&E). Myocarditis was graded as previously described [29] by examination of H&E-stained specimens at mid-ventricular cross sections. The grading using a 0−4 scale was recorded as follows: Grade 0, no inflammation; Grade 1, one to five distinct inflammatory foci with total involvement of 5% or less of the cross-sectional area; Grade 2, more than five distinct inflammatory foci, or involvement of more than 5% but less than 20% of the cross-sectional area; Grade 3, diffuse inflammation involving 20−50% of the area; Grade 4, diffuse inflammation involving more than 50% of the area.

### 2.5. Hemogram and Biochemistry

White blood cell (WBC) counts of the blood samples were analyzed by an automatic Coulter counter (HEMAVET HV950; Drew Scientific, Inc., Dallas, TX, USA). The plasma levels of alanine aminotransferase (ALT), creatinine (CRE), and creatine kinase (CK) were measured with the colorimetric method (Fuji Dri-Chem Slide, Fuji, Japan), according to instructions given by the manufacturer.

### 2.6. Flow Cytometry Analysis

Lung metastatic colon cancer-bearing mice were euthanized by intramuscular injection of a mixture of ketamine (100 mg/kg) and xylazine (10 mg/kg). The whole spleen, lung, and heart tissues were removed en bloc, and digested with solution containing collagenase A (1.5 mg/mL) and DNase I (0.4 mg/mL) at 37 °C for 30 min [30]. Cells were mashed through a 70 μm cell strainer to obtain single-cell suspensions, and further incubated with ACK (Ammonium-Chloride-Potassium) solution to lyse red blood cells. Before staining with cell surface markers, cells were incubated with an Fc receptor block (1 μg/1 × 10^6^ cells; BD Bioscience, San Diego, CA, USA) to reduce non-specific binding. Then, cells (1 × 10^6^ cells/mL for spleen, 1 × 10^7^ cells/mL for heart and lung) were stained with antibodies conjugated with the indicated fluorochromes for 20 min on ice, including anti-CD3-Alexa488, anti-NKG2D-PE, anti-PD-L1-PE/Dazzle 594, anti-Ly6G-PE/Cy7, anti-PD-1-APC, anti-F4/80-APC/Cy7, anti-Ly6C-BV421, anti-CD45-BV510, anti-CD11b-BV605, anti-CD4-BV650, and anti-CD8-BV785 antibodies (BioLegend, San Diego, CA, USA). After washing, cells were immediately analyzed on the CytoFLEX 13-color cytometer (Beckman Coulter, Brea, CA, USA) and quantified using CytExpert analysis software (Beckman Coulter, Brea, CA, USA). Immune cell populations were defined as the following: T cells (CD3^+^/CD11b^−^), CD8^+^ T cells (CD3^+^/CD11b^−^/CD8^+^), CD4^+^ T cells (CD3^+^/CD11b^−^/CD4^+^), NKG2D^+^ T cells (CD3^+^/CD11b^−^/NKG2D^+^), neutrophils (CD11b^+^/Ly6G^+^), macrophages (CD3^−^/CD11b^+^/Ly6C^+^/F4/80^+^), NK cells (CD3^−^/Ly6G^−^/CD11b^−^/NKG2D^+^), and inflammation monocytes (CD11b^+^/Ly6C^++^).

### 2.7. Statistical Analysis

Results were expressed as mean ± standard error of the mean (SEM). Statistical comparison in each experiment was performed using Student’s *t*-test or one-way analysis of variance (ANOVA). The difference was considered significant at *p* < 0.05. We used SigmaPlot version 8.0 (IBM SPSS, Armonk, NY, USA) with written syntax.

## 3. Results

### 3.1. Patient and Treatment

A 61-year-old woman with lung adenocarcinoma was sent to the emergency department with dyspnea and fatigue, three days after receiving her first dose of atezolizumab (1000 mg). Ten weeks before atezolizumab administration, she received five biweekly doses of nivolumab (3 mg/kg) with whole brain radiotherapy (RT) (30 Gy in 10 fractions) for brain metastasis, delivered after the first dose of nivolumab. Due to enlargement of lung nodules, shifting anti-PD-1 nivolumab to anti-PD-L1 atezolizumab was discussed in an oncology team meeting (Figure 1A). At day 3 of atezolizumab administration, the chest X-ray revealed right lung consolidation without fever and choking, which was not evident before atezolizumab administration (Figure 1B). Under the impression of pneumonitis, the dyspnea and lung consolidation subsided one week after treatment, which included high-dose methylprednisolone (5 mg/kg/day). However, the chest tightness and dyspnea developed four weeks later (day 40 of atezolizumab administration). The workup revealed sinus tachycardia by electrocardiography (Figure 1C), a normal troponin I level (<0.01 ng/mL, normal level <0.1), an elevated creatine kinase-myocardial band (CK-MB) level (10 ng/mL, normal level <7.2 ng/mL), and an elevated N-terminal pro-brain natriuretic peptide (NT-proBNP) level (2960 ng/mL, normal level <300 ng/mL) (Table 1). An echocardiogram revealed normal cardiac size and preserved global contractility of the left ventricle with an ejection fraction of 66.3% and no regional wall motion abnormality. Under a highly suspected diagnosis of myocarditis, she was treated with intravenous methylprednisolone at 5 mg/kg/day, and oral mycophenolate mofetil at 1000 mg/day. The progressive clinical deterioration was noted with serial elevation of troponin I, CK-MB, and NT-proBNP levels up to 1.3 ng/mL, 24 ng/mL, and 15,738 ng/mL, respectively. A subsequent echocardiogram revealed a non-significant decline of the ejection fraction to 59.2%. Cardiac arrest was noted at day 68 of atezolizumab administration, the 28th day after the development of cardiac symptoms.

### 3.2. Assessment of Cardiotoxicity after Combined ICI Therapies in Lung Metastasis Animal Model

To reveal the cardiotoxicity risk arising from ICI treatment, lung metastatic colon cancer-bearing mice were established using the intravenous injection method and were treated with the scheme simulating the treatment course of the aforementioned patient (Figure 2A). After three administrations of anti-PD-1/anti-PD-L1 antibodies, with either simultaneous or sequential treatment, heart tissues were collected for pathology and flow cytometry analysis. Hematoxylin and eosin staining showed no myocarditis lesions were observed in mice treated with anti-PD-1 or anti-PD-L1 alone (Figure 2B). However, in mice with the combination of anti-PD-1 and anti-PD-L1, either sequentially or simultaneously administered, myocarditis lesions equal or greater than grade 3 were noted. The characteristic features included extensive myocyte injury and patchy mononuclear infiltrates in the myocardium (Figure 2B, upper panel). Heart size in all groups had no marked alteration. Immunohistochemistry analysis showed Ly6C-positive staining was higher in the groups of PD-L1 inhibitor alone and concurrent PD-1 and PD-L1 blockade, whereas a Ly6G-positive signal appeared in the groups of blockade of PD-L1 alone, PD-1 plus PD-L1, and PD-1 followed with PD-L1 (Figure 2B, lower panel). Plasma levels of creatine kinase (CK) on 5 days, 10 days and 15 days post-treatment were further detected after ICI treatment. On day 10 post-treatment, plasma CK levels were statistically higher in the group of PD-L1 alone, concurrent and sequential treatment of PD-1 and PD-L1 inhibitors. On day 15 post-treatment, treatment with sequential blockade of PD-1 and PD-L1 caused a higher level of plasma CK but this was not statistically significant (Figure 2C). An abnormal infiltration of leukocytes within hearts were further examined by flow cytometry, and a higher level of inflammatory monocytes (CD11b^+^/Ly6C^++^) were detected in the hearts of mice treated with anti-PD-1 plus anti-PD-L1 antibody (12.85% ± 3.38%) than those of control group mice (2.80% ± 0.36%) (Figure 2D,E). Sequential administration of PD-1 and PD-L1 inhibitors in lung metastasis mice showed an increase of neutrophils (17.01% ± 0.12%) rather than inflammatory monocytes, as compared to control mice (4.49% ± 2.43%). Besides, macrophages in the heart also decreased after sequential treatment of PD-1 and PD-L1 inhibitors (51.42% ± 3.25%) than untreated mice (59.67% ± 2.69%). Intriguingly, when treatment with the PD-L1 inhibitor was followed with a PD-1 blocker (α-PD-L1, α-PD-1), accumulation of inflammatory monocytes or neutrophils was not observed in murine heart tissues (Figure 2D,E). Other immune cells in the hearts did not show significant changes after combined ICI treatment, including T cells, dendritic cells (DCs), and NK cells. 

### 3.3. Assessment of Tumor Control after Combined ICI Therapies in Lung Metastasis Animal Model

In order to determine whether different courses of ICI treatment can lead to different adverse effects and tumor control, biological toxicity and survival analyses were performed. IVIS images showed that combinatorial treatments exhibited better tumor control than monotherapy of anti-PD-1 or anti-PD-L1 in lung metastatic colon cancer-bearing mice (Figure 3A). Among combinatorial treatments, lung metastatic mice treated with anti-PD-L1 following anti-PD-1 presented the lowest signal of IVIS images, indicating tumor growth was successfully attenuated with sequential treatment of anti-PD-L1 and anti-PD-1 therapy (Figure 3A). No significant changes in body weight and renal function were noted within 15 days post-treatment in any mouse group (Figure 3B,E). Consistent and significant suppression of white blood cells were observed on day 5 and day 10 in all ICI therapies (Figure 3C). Among the combined ICI treatment groups, only sequential administration of PD-1 and PD-L1 inhibitors generated a transient increase of alanine aminotransferase (ALT), an indicator of liver function, on day 10 post-treatment (*p* = 0.09, 3/6 mice showed abnormal ALT). Survival analysis was revealed, and ICI treatment was started on day 7 of the designed scheme to make sure that the lung metastatic tumor cells were undergoing exponential growth. Among all treatment groups, the anti-PD-L1 followed by anti-PD-1 was the most effective combination protocol to control lung metastasis from CT26 colon cancer cells (*p* = 0.03) (Figure 3F). Monotherapy of anti-PD-L1 also showed improvement of overall survival compared to control mice (*p* = 0.05), even though tumor volumes were big. These data indicate that in the lung metastatic mouse model, ICI-mediated infiltration of immune cells into the myocardium did not cause severe lethality but showed better survival benefit.

### 3.4. Alternation of Circulating Immune Cells in Lung Metastatic Mice after Combined ICI Therapies

First, the effectiveness of ICI treatment on the expression of PD-1 and PD-L1 in a lung metastatic colon cancer mouse model was revealed on day 10 by flow cytometric analysis. Our data indicated that PD-1 blockade caused prolonged and effective inhibition of PD-1 expression in both leukocytes and non-leukocytes (including tumor cells) in the lung (Figure 4A). In contrast, PD-L1 blockade presented with a more transient suppression of PD-L1, as the level of PD-L1 was lower in the group of anti-PD-1 followed with anti-PD-L1 inhibitors, but recovered to normal levels in the anti-PD-L1 alone group (Figure 4A). Combinatorial blockade exhibited various effects on PD-1 and PD-L1 expression in lung metastatic mice. Intriguingly, only the sequential combination of anti-PD-L1 followed by anti-PD-1 (the most effective regimen for controlling lung metastasis) universally inhibited the up-regulated expression of both PD-1 and PD-L1 in lung metastasis (Figure 4A). It was worth noting that both anti-PD-1 and anti-PD-L1 monotherapy resulted in the elevation of PD-L1 expression in the tumor microenvironment seven days post-treatment. Besides, the expression of PD-1 and PD-L1 on immune cells in the spleen made them more susceptible to treatment with the anti-PD-L1 inhibitor (Figure 4A). Circulating immune cells such as neutrophils, inflammatory monocytes, macrophages, NK cells, and DCs were further examined in lung metastatic mice in response to ICI therapies. Lung metastasis of colon cancer led to increased neutrophils and decreased T cells in the spleen, and combinatorial and sequential treatments of PD-1/PD-L1 inhibitors reversed these alternations (Figure 4B). A consistent suppression of neutrophils was observed on day 17 post-treatment (Figure 4C). Macrophages in the spleen were significantly increased on day 17 after monotherapy and various combination treatments of PD-1 and PD-L1 inhibitors (Figure 4C).

### 3.5. Alternations of Local Immune Cells in Lung Metastatic Mice after Combined ICI Therapy

The immune cell profiles at the tumor site were assessed on day 17 post-treatment in lung metastatic colon cancer-bearing mice with or without ICI treatment. Flow cytometric analysis showed the alternation of T cells and neutrophils (Figure 5A) and the changes of monocytes and macrophage populations (Figure 5B) in response to ICI treatment. Compared to lung metastatic mice and normal mice, tumor cells that metastasized to the lungs caused a decrease in T cells, including CD8^+^ T cells and CD4^+^ T cells. This decrease in number was restored by a combination of PD-1 and PD-L1 inhibitors (both sequential and simultaneous), but not by each inhibitor alone (Figure 5E). Lung metastasis increased the numbers of local immune cells such as tumor associated-neutrophils, macrophages, and NK cells. Combination blockade (both sequential and simultaneous) subverted the increase in numbers to an extent greater than each inhibitor alone (Figure 5E). In addition to the comprehensive profile of immune cells, tumor-infiltrating leukocytes expressing PD-1 or PD-L1 were simultaneously examined. Among T cells in the tumor microenvironment, CD8^+^ T cells and NKG2D^+^ T cells were the major cells expressing PD-1 (Figure 5C) and PD-L1 (Figure 5D), respectively. Quantification analysis summarized that up-regulated expression of PD-1 in CD8^+^ T cells, NK cells, and macrophages—as well as PD-L1 in NKG2D^+^ T cells, macrophages, and neutrophils—was noted in lung metastatic mice compared to normal mice (Figure 5F,G). Combination blockade inhibited the expression of PD-1 in CD8^+^ T cells, NK cells, macrophages, and that of PD-L1 in NKG2D^+^ T cells, macrophages, and neutrophils (Figure 5F,G). 

## 4. Discussion

Cardiotoxicity induced by ICI therapy is an underestimated and emerging issue in clinical trials, especially for those with a combination design of ICIs. Myocarditis is life-threatening and difficult to detect early. So far, the pathogenesis of immune-related myocarditis remains unclear. For validating the finding from a lung adenocarcinoma patient, we applied a colon CT26 cancer with lung metastases model to test the effect of dual blockade in mice. The main reason for choosing this model is that it has been widely used in immune checkpoint blockade studies, including for the blockade for PD-1 and PD-L1 independently, with a more comprehensive understanding of dynamic alterations in immune profiles. After this proof-of-concept study, further investigations using various types of cancers are warranted. In our animal model, concurrent and sequential treatment of PD-1 and PD-L1-blocking antibodies resulted in infiltrating leukocyte accumulation in the heart, an abnormality that was not observed in mono-therapy groups. Intriguingly, the administration of the PD-L1 inhibitor prior to the PD-1 inhibitor did not cause leukocytic infiltration of the myocardium. PD-1 is a co-inhibitory molecule of the B7/CD28 superfamily, which can bind PD-L1 and PD-L2 to negatively regulate responses of immune cells including T cells, B cells, macrophages, and dendritic cells [31]. Genetic deletion of PD-1 has been reported to cause autoimmune myocarditis with dilated cardiomyopathy in mice [32]. This implies that PD-1 may play a role in myocardial immune responses and may protect against inflammation and myocyte damage in T-cell-mediated myocarditis. Tarrio et al. revealed transfer of ovalbumin-specific CD8^+^ T cells into cMyc-mOVA (cMyc oncogene and membrane-bound form of ovalbumin) mice resulted in enhanced immunization accompanied with more myocardial inflammation in recipients receiving PD-1 null T cells [33]. They concluded that PD-1 deficient T cells are more efficient killers of target cells and induce more inflammation, as evident with the increase of CD8^+^ T cells, neutrophils, and macrophages in murine myocardium. In our case, there is no significant induction of CD8^+^ T cells in the heart, and inflammatory monocytes (Ly6C^++^), but not macrophages (Ly6C^+^) are significantly increased on day 11 post-treatment of PD-1 plus PD-L1 blockade in the lung metastasis mouse model. Another case report that resembled our clinical case mentioned that PD-1 blockade-induced myocarditis was identified in a patient with lung squamous cell carcinoma who received simultaneous whole brain radiotherapy [34]. In our case, myocarditis suddenly appeared on day 3 post-treatment of PD-L1 blockade but not in the therapeutic period of the PD-1 inhibitor combined with radiotherapy, implying that the combinatorial use of PD-1 and PD-L1 inhibitors may lead to a substantial increase of immunotherapy-induced cardiomyopathy.

In the inflamed myocardia of patients with fatal myocarditis after a combination of CTLA-4 and PD-1 blockade, PD-L1 was expressed on the membranous surface of injured cardiac myocytes and on infiltrating CD8^+^ T cells and histiocytes of the inflamed myocardium, but not skeletal muscles. The mRNA expression data from another ICI-induced myocarditis patient showed 10-fold more abundant expression of PD-L1 in affected cardiac tissue, which was five-fold higher than in affected skeletal muscle [23]. Given that human and murine myocytes constitutively express PD-1 and PD-L1, and expression of PD-L1 is up-regulated in injured myocytes [35,36], the PD-1 blockade-injured myocytes may cause secondary PD-L1 up-regulation, aiming to attenuate pro-inflammatory reactions. This raises the possibility that concurrent or subsequent blockade of PD-L1 added to PD-1 treatment may block the salvage mechanism of anti-PD-1-injured myocytes, leading to the development of fulminant myocarditis. In our animal model, the level of PD-L1 in the myocardium is susceptible to anti-PD-L1 blockade treatment. Besides, a higher dosage (over 200 µg/mice per treatment) of anti-PD-L1 antibody led to significant lethality in our animal system, implying that the toxicity of anti-PD-L1 in vivo should be carefully manipulated while changing the dosage. A recent study reported synergistic toxicities (pneumonitis and colitis) were found only in patients treated with sequential PD-(L)1 blockade and an EGFR (epidermal growth factor receptor) tyrosine kinase inhibitor, osimertinib [37]. Based on their observation, receptor occupancy of the anti-PD-(L)1 antibody is longer than that of osimertinib; thus, there was evident toxicity in lung metastatic mice treated with PD-1 followed by PD-L1 inhibitors. Furthermore, myocarditis was also reported in a patient with multiple myeloma who was treated with an immunomodulatory drug (lenalidomide-dexamethasone) combined with an anti-PD-1 inhibitor (pembrolizumab) [38]. Autopsy revealed the increase of infiltrating macrophages, CD8^+^ T cells and focal fibrosis in the myocardium after combination therapy. A better understanding of the mechanism of this ICI-induced cardiac toxicity may provide insight into the development of preventive or therapeutic agents for immune-related cardiotoxicity.

The infiltrating CD68^+^ macrophages in ICI-induced myocarditis were noted in a patient with fatal myocarditis after a combination of CTLA-4 and PD-1 blockade. PD-1 expression in M2 macrophages in tumors is reported to be associated with disease progression and impaired phagocytotic potency against tumors in the same animal model used in this study [39]. In our animal system, combination blockade of PD-1 and PD-L1 reduced the expression of PD-1 and PD-L1 in macrophages, which may imply the validation of the blockade effect but, on the other hand, may represent the relative activation of macrophage function. Besides, transient elevation of neutrophils in the heart and abnormal liver function were only observed on day 10 of the sequential treatment of PD-1 and PD-L1 blockade, indicating the ability of animals to recover differs with clinical results in response to ICI therapy. Moreover, macrophages and neutrophils are the major immune cells expressing PD-L1 in the tumor microenvironment; thus, the inhibition of PD-L1 may influence the activities and functions of tumor-associated macrophages and neutrophils. It is worth noting that a subset of T cells, NKG2D^+^ T cells, also expressed PD-L1. The NKG2D receptor has received great attention in the development of novel therapeutic agents, and was found to be expressed on NK cells and T cells in both humans and mice [40,41]. For T cells, both NKG2D and CD28 function as costimulatory receptors for CD8^+^ T cell memory formation, and the ligands of NKG2D are broadly expressed in many cell types upon stress stimuli [42]. Our data indicate that combination blockade could reverse the tumor-associated alterations in immune cell lineages, including the increase of CD8^+^ T cells and NKG2D^+^ T cells. Whether this immunomodulatory effect is correlated to toxicity and greater tumor control by combination blockade remains to be elucidated.

## 5. Conclusions

The combinatory use of PD-1 and PD-L1 blockade, either sequentially or concurrently, may cause fulminant cardiotoxicity, and such usage should be cautionary. The combinatorial treatment of anti-PD-L1 followed by anti-PD-1 is more effective than other mono- or combined strategy.

## Figures and Tables

**Figure 1 cancers-11-00580-f001:**
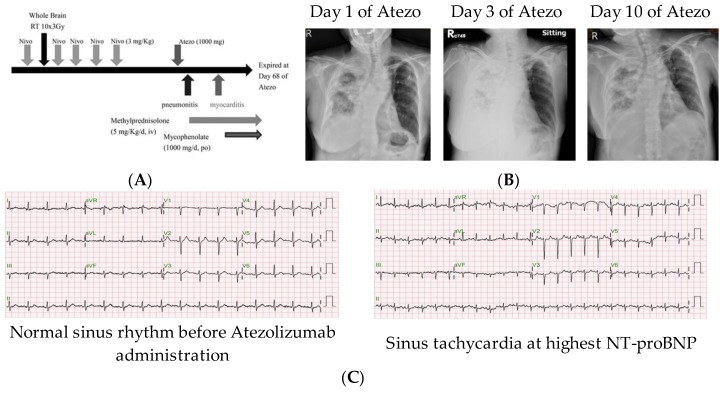
Clinical course of the presented patient. (**A**) Scheme of immune checkpoint inhibitor treatment course for the patient with brain metastatic lung adenocarcinoma. (**B**) Chest X-ray films demonstrated Atezolizumab (Atezo)-associated pneumonitis. (**C**) Electrocardiography of the patient before and after Atezolizumab administration. Nivo = nivolumab; RT = radiotherapy; NT-proBNP = N-terminal pro-brain natriuretic peptide.

**Figure 2 cancers-11-00580-f002:**
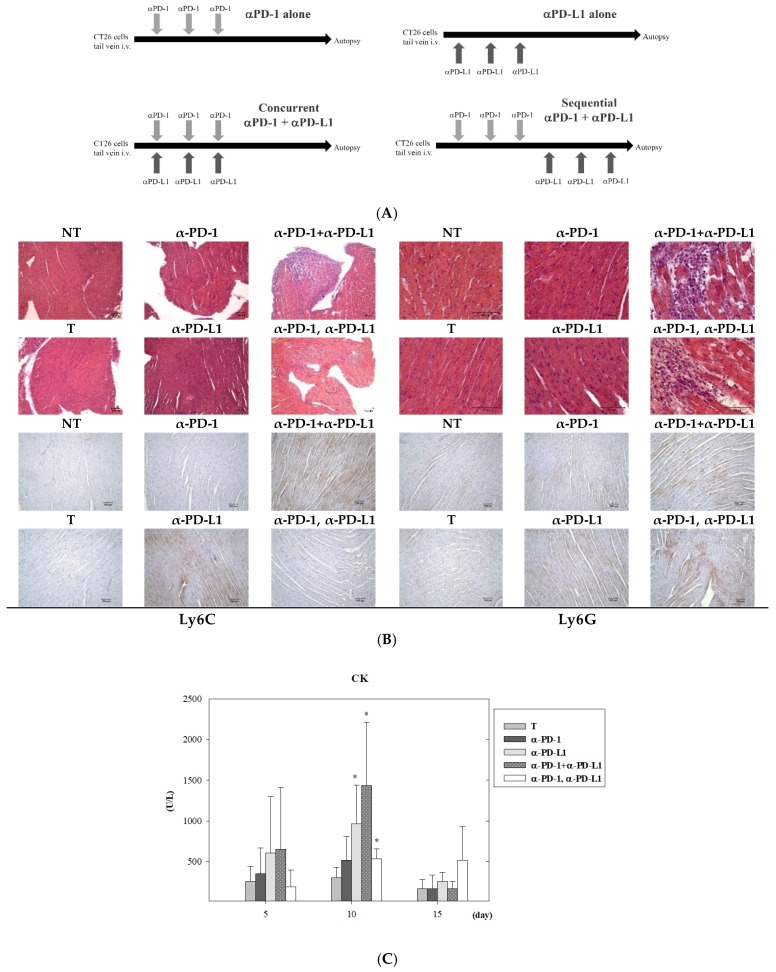
Combination treatment of programmed cell death 1/programmed cell death ligand 1 (PD-1/PD-L1) blockade induced myocarditis in a lung metastatic colon cancer mouse model. (**A**) Scheme of experimental design. Balb/c mice were intravenously (i.v.) injected with CT26-luciferase expressing cells. Single and combined treatment of anti-PD-1 and anti-PD-L1 were administered intraperitoneally (i.p.) on days 0, 2, and 4. Sequential treatments were administered on days 0, 2, and 4, followed by another antibody on days 6, 8, and 10. Mice were sacrificed for analysis on day 11. (**B**) Representative hematoxylin and eosin-stained (H&E) slides of heart tissues from each group (left panel: 200×; right panel: 400×). Lower panel shows the immunohistochemistry (IHC) stain of Ly6C and Ly6G expression in heart sections at 100× magnification. Scale bars indicate 100 μm. (**C**) Plasma levels of creatine kinase (CK) on 5 days, 10 days and 15 days post-treatment after immune checkpoint inhibitor (ICI) therapies. (**D**) Flow cytometric analysis of immune cell profiles from heart tissues revealed an increase of inflammatory monocytes (Ly6C^++^) and neutrophils (Ly6G^+^) in lung metastatic mice treated with concurrent and sequential PD-1/PD-L1 inhibitors, respectively. (**E**) Quantification results from flow cytometric data indicated that administration of anti-PD-1 followed with anti-PD-L1 antibodies led to the accumulation of neutrophils in the hearts of lung metastatic mice. NT and T indicate the no tumor and tumor group, respectively. In ICI-treated groups, statistical analysis was performed with the tumor (T) group (* *p* < 0.05). Total leukocytes (CD45^+^) = 100%. *n* = 3 for each group.

**Figure 3 cancers-11-00580-f003:**
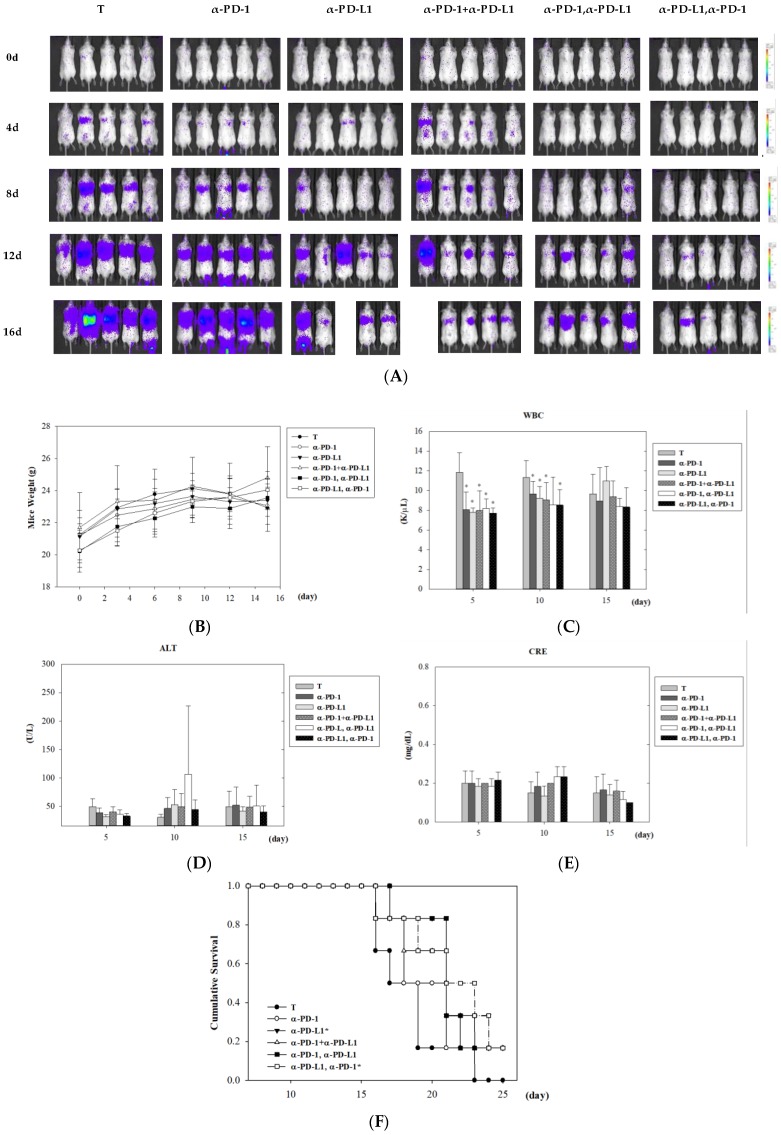
Therapeutic effects and toxicity in a lung metastatic animal model of monotherapy and various combination treatments of anti-PD-1 and anti-PD-L1. (**A**) Tumor growth of lung metastatic colon cancer-bearing mice after mono-, combined, and sequential treatment of PD-1 and PD-L1 antibodies. After seven days of intravenous cell injection (day 0), mice were treated with the indicated ICI therapies. Tumor growth of lung metastatic colon cancer was monitored using the IVIS imaging system at four-day intervals. General toxicities were examined by mouse weight (**B**), white blood cells (**C**), liver function with ALT (**D**), and renal function with CRE (**E**) within 15 days post-treatment. (**F**) Survival analysis was assessed in lung metastatic mice with visible tumor burden in the lung. In ICI-treated groups, statistical analysis was performed with the tumor (T) group (* *p* < 0.05). *n* = 6 for each group. ALT = alanine aminotransferase; CRE = creatinine.

**Figure 4 cancers-11-00580-f004:**
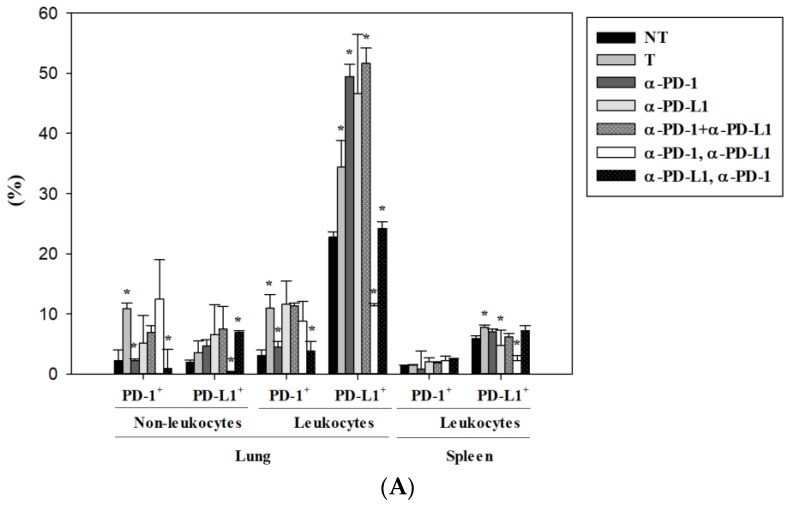
Alterations of circulating immune cells in a lung metastatic animal model after monotherapy and various combination treatments of anti-PD-1 and anti-PD-L1. (**A**) Flow cytometric analysis for expression of PD-1 and PD-L1 on leukocytes (CD45^+^) and non-leukocytes (CD45^−^) in the lung and spleen on day 11 post-treatment. (**B**) Immune cell composition in the spleen on day 11 post-treatment in lung metastasis mice. (**C**) Immune cell composition in the spleen on day 17 post-treatment in lung metastasis mice. The tumor (T) group from lung metastatic mice was compared with the no tumor (NT) group from normal mice. In ICI-treated groups, statistical analysis was performed with the tumor (T) group (* *p* < 0.05). Total leukocytes (CD45^+^) = 100%. Immune cells were defined as follows: neutrophils (CD11b^+^/Ly6G^+^), inflammatory monocytes (CD11b^+^/Ly6C^++^), macrophages (CD3^−^/CD11b^+^/Ly6C^+^/F4/80^+^), T cells (CD3^+^/CD11b^−^). *n* = 3 for each group.

**Figure 5 cancers-11-00580-f005:**
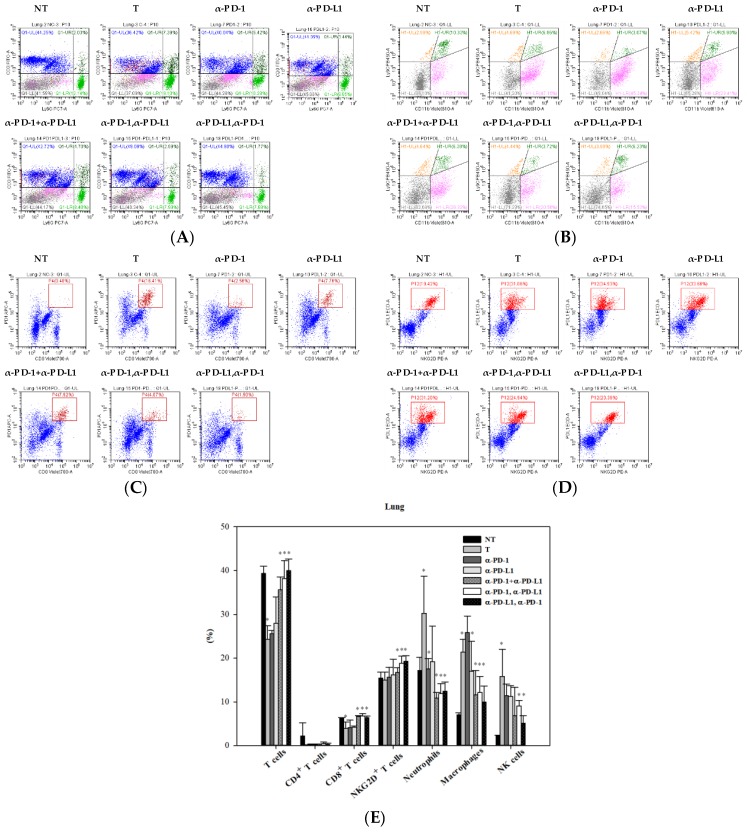
Alterations of local immune cells in a lung metastatic animal model after monotherapy and various combination treatments of anti-PD-1 and anti-PD-L1. (**A**) Dot plots of flow cytometric analysis revealed the alteration of T cells (upper left region in blue) and neutrophils (upper and lower right region in green) on day 17 post-treatment. (**B**) Other immune cells with double negative CD3 and Ly6G (lower left region of **A**) were further gated and analyzed to distinguish macrophages/monocytes/DCs (lower right, pink color) and inflammatory monocytes (upper right, dark green color) by Ly6C and CD11b staining. (**C**) The proportion of PD-1-positive cells in CD8^+^ T cells in response to ICI therapies. (**D**) The proportion of PD-L1 positive cells in NKG2D^+^ T cells in response to ICI treatment. (**E**) Quantification of the immune cell profile in the lung on day 17 post-treatment in lung metastasis mice. (**F**) Quantification of major immune cells expressing PD-1 in lung metastases on day 17 post-treatment. (**G**) Quantification of major immune cells expressing PD-L1 in lung metastases on day 17 post-treatment. The tumor (T) group from lung metastatic mice was compared with the no tumor (NT) group from normal mice. In ICI-treated groups, statistical analysis was performed with the tumor (T) group (* *p* < 0.05). Total leukocytes (CD45^+^) = 100%. Immune cells were defined as follows: T cells (CD3^+^/CD11b^−^), CD8^+^ T cells (CD3^+^/CD11b^−^/CD8^+^), CD4^+^ T cells (CD3^+^/CD11b^−^/CD4^+^), NKG2D^+^ T cells (CD3^+^/CD11b^−^/NKG2D^+^), neutrophils (CD11b^+^/Ly6G^+^), macrophages (CD3^−^/CD11b^+^/Ly6C^+^/F4/80^+^), NK cells (CD3^−^/Ly6G^−^/CD11b^−^/NKG2D^+^). *n* = 3 for each group.

**Table 1 cancers-11-00580-t001:** Data from echocardiogram and serum cardiac enzyme levels.

Cardiac Function Parameters	d40 of Atezo	d45 of Atezo	d55 of Atezo
**LV ejection fraction**	66.3%	N.A.	59.2%
**Troponin I (ng/mL)**	<0.01	0.3	1.3
**CK-MB (ng/mL)**	10	27	24
**NT-proBNP (ng/mL)**	2960	8668	15,738

d = day; LV = left ventricular; CK-MB = creatine kinase-myocardial band; NT-proBNP = N-terminal pro-brain natriuretic peptide.

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
