# Peer review of "Sequential Blockade of PD-1 and PD-L1 Causes Fulminant Cardiotoxicity—From Case Report to Mouse Model Validation"

_cancers, 2019, doi:10.3390/cancers11040580_

Round 1

Reviewer 1 Report

The manuscript “Sequential Blockade of PD-1 and PD-L1 Causes Fulminant Cardiotoxicity—Mouse model to validate clinical observation” authored by Shin-Yi Liu, Wen-Chien Huang, et al. is an interesting research. They investigate the risk of combined administration of PD-1 and PD-L1 inhibitors for inducing myocarditis on cancer patients. To prove the hypothesis, they also proved the results on mice model. The project is good fitted to the scope of CANCERS. And I believe its results could benefit the readers in the fields of cancers, especially the clinical applications. I am suggesting to accept the manuscript with small edition. The evidence on which my decision was made was shown following:

The PD-1 pathway provides an important circuit for immune regulation, with the PD-1 receptor expressed on T cells, and the PD-L1 ligand expressed in multiple tissues and cell types. The PD-L1 functions to provide a level of immune inhibitor to prevent uncontrolled, destructive, T-cell-mediated response and autoimmunity. It is always a popular cure strategy to use PD-1 and PD-L1 in combination or in sequence for the cancer. But the side effect or toxicity of these combinate using is unknown. This project studied the related toxicity of combination using of PD-1 and PD-L1 since they found a patient with fatal myocarditis after the PD-1 and PD-L1 inhibitors. But the problem is: only one patient situation is not enough to prove the hypothesis. The N number is too small, and there might be some other reasons to induce the toxicity. There is one more publication this year (Annals of Oncology, mdz077, https://doi.org/10.1093/annonc/mdz077

), which showed the potentially toxic between PD-(L)1 and osimertinib, but not with erlotinib on the EGFR mutant NSCLC. Their patient number is 126, which is big enough to get the conclusion.

Although the authors tested the hypothesis on mice, it is still urge demand to collect more clinical data to prove the conclusion, since it will take a long-long time to get clinical outcomes from animal results. In addition, the irAE is usually on EGFR mutant cancer. In this paper, the patient information only indicated metastatic lung adenocarcinoma. What is the subtype of the lung cancer, (NSCLC, or SCLC, or LCT) and the toxicity may be only on the specific subtype.

For the experiment design on mice, why there is no group of (PD-L1 first+PD-1 later). The pathways of PD-1 and PD-L1 is different, it is reasonable to test the different sequentially order. But it seems the results has the results of (PD-L1 first+PD-1 later).

Basically, I think this is a good study, especially for the clinical application. More patients data will be better, although I know it is not easy.

Author Response

Response to Reviewer 1 Comments

Reviewer 1:

The manuscript “Sequential Blockade of PD-1 and PD-L1 Causes Fulminant Cardiotoxicity—Mouse model to validate clinical observation” authored by Shin-Yi Liu, Wen-Chien Huang, et al. is an interesting research. They investigate the risk of combined administration of PD-1 and PD-L1 inhibitors for inducing myocarditis on cancer patients. To prove the hypothesis, they also proved the results on mice model. The project is good fitted to the scope of CANCERS. And I believe its results could benefit the readers in the fields of cancers, especially the clinical applications. I am suggesting to accept the manuscript with small edition. The evidence on which my decision was made was shown following:

Point 1: The PD-1 pathway provides an important circuit for immune regulation, with the PD-1 receptor expressed on T cells, and the PD-L1 ligand expressed in multiple tissues and cell types. The PD-L1 functions to provide a level of immune inhibitor to prevent uncontrolled, destructive, T-cell-mediated response and autoimmunity. It is always a popular cure strategy to use PD-1 and PD-L1 in combination or in sequence for the cancer. But the side effect or toxicity of these combinate using is unknown. This project studied the related toxicity of combination using of PD-1 and PD-L1 since they found a patient with fatal myocarditis after the PD-1 and PD-L1 inhibitors. But the problem is: only one patient situation is not enough to prove the hypothesis. The N number is too small, and there might be some other reasons to induce the toxicity. There is one more publication this year (Annals of Oncology, mdz077, https://doi.org/10.1093/annonc/mdz077), which showed the potentially toxic between PD-(L)1 and osimertinib, but not with erlotinib on the EGFR mutant NSCLC. Their patient number is 126, which is big enough to get the conclusion.

Response 1: We initiated the mice study after clinical observation of a patient received dual blockade of PD-1 and PD-L1 and subsequently developed lethal myocarditis. We would like to categorize the current work as a translational research with clinical insight. The main reason for the small test sample number is that we attempted to validate the unexpected finding by animal model to avoid possible harmful test in patients. After this proof-of-concept study, the further validation in clinical investigation could be considered and should be performed very carefully. The reference reviewer mentioned has been described in the Discussion (Line 470 and ref. 37) as follows:

“Recent study reported synergistic toxicities (pneumonitis and colitis) were found only in patients treated with sequential PD-(L)1 blockade and EGFR tyrosine kinase inhibitor, osimertinib [37]. Based on their observation, receptor occupancy of anti-PD-(L)1 antibody is longer than that of osimertinib. Similar to our notice, receptor occupancy of anti-PD-1 antibody is longer than that of anti-PD-L1; thus, there was evident toxicity in lung metastatic mice treated with PD-1 followed by PD-L1 inhibitors.”

Comment 2: Although the authors tested the hypothesis on mice, it is still urge demand to collect more clinical data to prove the conclusion, since it will take a long-long time to get clinical outcomes from animal results. In addition, the irAE is usually on EGFR mutant cancer. In this paper, the patient information only indicated metastatic lung adenocarcinoma. What is the subtype of the lung cancer, (NSCLC, or SCLC, or LCT) and the toxicity may be only on the specific subtype.

Response 2: The subtype of lung cancer of this patient is adenocarcinoma. This description could be seen in the Abstract (Line 17) and Results (Line 140).

Comment 3: For the experiment design on mice, why there is no group of (PD-L1 first+PD-1 later). The pathways of PD-1 and PD-L1 is different, it is reasonable to test the different sequentially order. But it seems the results has the results of (PD-L1 first+PD-1 later). 

Response 3: We appreciate the reviewer’s question. We examined cardiac toxicity and collected murine heart specimen for HE staining under the design of sequential use of PD-1 followed with PD-L1 inhibitors. For the following study including comprehensive immune cells profile and therapeutic effect, reverse design of PD-L1 followed with PD-1 inhibitors was added and presented unexpected results. We re-collected murine heart tissues after ICI therapies for flow cytometric analysis and found PD-L1 followed with PD-1 blockade did not showed any increase of inflammatory monocytes and neutrophils at day 11 post-treatment.

Comment 4: Basically, I think this is a good study, especially for the clinical application. More patients data will be better, although I know it is not easy.

Response 4: After this proof-of-concept study, the further prospective validation in clinical investigation could be considered and should be performed very carefully. We will try to retrospectively collect more patients received dual blockade.

Reviewer 2 Report

The authors have pointed out that th therapy with immunecheck point blockers can lead to myocarditis. This is shown in a murine experimental model of lung metastasis of the colon rectal carcinoma cell line CT26 and in a patient with lung carcinoma. It appears that the treatment in sequence with anti-PD1 and anti-PDL1 antibodies can have this effect.

The paper is of interest and complexively well written. This information is not completely novel but it is of interest that the specific combination of antibodies is effective limiting the tumor growth and inducing myocarditis.

There are some points that should be addressed

The descrption of the clinical features of the patient is not complete. Indeed, it is not indicate why this specific patient have been treated with radiotherapy to brain. One can imagine that lung tumor mestastasis could be present in the brain but this point is not clarified by the Authors.

The figure 1 panel C is not correct. Indeed, looking at the heart frequency, it is evident that the upper electrocardiogram gives a frequency of about 150b/min while the lower is about 100b/min . In the text, it has been stated the opposite as indicated in the figure. This should be clearly explained because it indicates that no care has been taken by the Authors at this key point of the paper.

The infiltration of neutrophils (in particular) as well as of macrophages should be shown by immunohistochemistry to reinforce the data obtained with cytofluorimetry.

The presence of serum markers of myocarditis in mice treated with immune check point blockers should be shown to reinforce the idea that indeed the increment of neutrophils can lead to this inflammation process. It is not clear what the condition indicated with “C” and “NC” represent. It is evident that these are controls (at least I think) but there is no explanation in the text and the results obtained are difficult to interpret if this is not clarified.

The colours used to show the percentages of the different leukocyte cell subsets are not really fine (different levels of gray). In some instances different experimental conditions are almost identical and also for this reason some results are difficult to be interpreted.

Several (if not all) immunocytofluorimetric results are shown as graphs/column. Nor FACS plots neither negative controls of dot plots are shown. This does not allow the reader (and the reviewer) to correctly evaluate the significance of data.

Furthermore, it is not clear why the authors have analyzed the expression of PD1 and PDL-1 on leukocytes in great detail, while no evident expression of PDL-1 is shown for CT26 cell line. Indeed, one can hypothesize that the therapy with anti-PDL-1 in the mouse model can be efficient if this marker is expressed on tumor cells beside leukocytes. Viceversa for the expression of PD1 which should inhibit immune cell response, thus it is relevant to know if this marker is present on lymphocytes and how it is modulated during the therapy.

In addition, I would say that the model used consisting of colon cancer cells localized in the lung is not really reminescent of the case report illustrated at the beginning of this paper (a lung adenocarcinoma). This should be pointed out because it is a limitation to transfer the information given to a clinical setting. Finally, it appears that some relevant references on the topic are missing. Indeed, the works which follow should be cited and discussed.

 1)      Anti-PD1 associated fulminant myocarditis after a single pembrolizumab dose: the role of occult pre-existing autoimmunity. Martinez-Calle N, Rodriguez-Otero P, Villar S, Mejías L, Melero I, Prosper F, Marinello P, Paiva B, Idoate M, San-Miguel J. Haematologica. 2018 Jul;103(7):e318-e321. doi: 10.3324/haematol.2017.185777. Epub 2018 Apr 12. No abstract available

2)      PD-1 protects against inflammation and myocyte damage in T cell-mediated myocarditis. Tarrio ML, Grabie N, Bu DX, Sharpe AH, Lichtman AH. J Immunol. 2012 May 15;188(10):4876-84. doi: 10.4049/jimmunol.1200389. Epub 2012 Apr 9. PMID:22491251

3)      Drug-induced myocarditis after nivolumab treatment in a patient with PDL1- negative squamous cell carcinoma of the lung. Semper H, Muehlberg F, Schulz-Menger J, Allewelt M, Grohé C. Lung Cancer. 2016 Sep;99:117-9. doi: 10.1016/j.lungcan.2016.06.025. Epub 2016 Jul 1. PMID:27565924

Author Response

Response to Reviewer 2 Comments

Reviewer 2:

The authors have pointed out that therapy with immune checkpoint blockers can lead to myocarditis. This is shown in a murine experimental model of lung metastasis of the colon rectal carcinoma cell line CT26 and in a patient with lung carcinoma. It appears that the treatment in sequence with anti-PD1 and anti-PDL1 antibodies can have this effect.

 The paper is of interest and complexively well written. This information is not completely novel but it is of interest that the specific combination of antibodies is effective limiting the tumor growth and inducing myocarditis.

 There are some points that should be addressed

Comment 1: The description of the clinical features of the patient is not complete. Indeed, it is not indicate why this specific patient have been treated with radiotherapy to brain. One can imagine that lung tumor mestastasis could be present in the brain but this point is not clarified by the Authors.

Response 1: We appreciate the helpful comment to point out our mistake. We have added a description for the reason of brain RT in the Results (Line 143) as follows:

Ten weeks before atezolizumab administration, she received 5 biweekly doses of nivolumab (3 mg/Kg) with whole brain RT (30 Gy in 10 fractions) for brain metastasis, delivered after the first dose of nivolumab.

Comment 2: The figure 1 panel C is not correct. Indeed, looking at the heart frequency, it is evident that the upper electrocardiogram gives a frequency of about 150b/min while the lower is about 100b/min. In the text, it has been stated the opposite as indicated in the figure. This should be clearly explained because it indicates that no care has been taken by the Authors at this key point of the paper.

Response 2: Thank you for pointing out the mistake. We did put the EKG figures in opposite position. We have corrected this editing error in revised Figure 1C.

Comment 3: The infiltration of neutrophils (in particular) as well as of macrophages should be shown by immunohistochemistry to reinforce the data obtained with cytofluorimetry.

Response 3: To validate the findings about infiltration of neutrophils and macrophages, we performed additional experiments by using immunohistochemistry. Consistent to flow cytometric analysis, Ly6C-positive cells are highest in PD-1+PD-L1 group, and Ly6G-positive cells are highest in PD-1 followed with PD-L1 group. The new data have been added in Figure 2B and related descriptions in the Experimental Section (Line 91), Results (Line 189) as follows:

“Immunohistochemistry analysis showed Ly6C-positive staining was higher in the groups of PD-L1 inhibitor alone and concurrent PD-1 and PD-L1 blockade, whereas Ly6G-positive signal appeared in the groups of blockade of PD-L1 alone, PD-1 plus PD-L1, and PD-1 followed with PD-L1 (Figure 2B, lower panel).”

Comment 4: The presence of serum markers of myocarditis in mice treated with immune check point blockers should be shown to reinforce the idea that indeed the increment of neutrophils can lead to this inflammation process. 

Response 4: The serum markers for myocardial injury in mice have been added in revised Figure 2C and the related descriptions have been added in the in Experimental Section (Line 112), Results (Line 192) as follows:

“Plasma levels of creatine kinase (CK) on 5, 10, 15 days post-treatment were further detected after ICI treatment. On day 10 post-treatment, plasma CK levels were statistically higher in the group of PD-L1 alone, concurrent and sequential treatment of PD-1 and PD-L1 inhibitors. On day 15 post-treatment, treatment of sequential blockade of PD-1 and PD-L1 kept higher level of plasma CK but not statistically significant (Figure 2C).”

Comment 5: It is not clear what the condition indicated with “C” and “NC” represent. It is evident that these are controls (at least I think) but there is no explanation in the text and the results obtained are difficult to interpret if this is not clarified.

Response 5: The “NC” and “C” stand for “Negative Control, no tumor” and “Control, tumor implanted without treatment”, respectively. To avoid confusion, we have renamed “NC” and “C” to be “NT” for no tumor and “T” for untreated tumor in the entire manuscript, respectively.

Comment 6: The colours used to show the percentages of the different leukocyte cell subsets are not really fine (different levels of gray). In some instances different experimental conditions are almost identical and also for this reason some results are difficult to be interpreted.

Response 6: To improve the quality of figures, we have changed the different levels of gray to different styles for bars of all revised figures.

Comment 7: Several (if not all) immunocytofluorimetric results are shown as graphs/column. Nor FACS plots neither negative controls of dot plots are shown. This does not allow the reader (and the reviewer) to correctly evaluate the significance of data.

Response 7: The representative dot plots of flow cytometry data have been added in revised Figure 5A-D and described in the Results (Line 367-378) as follows:

“Flow cytometric analysis showed the alternation of T cells and neutrophils (Figure 5A) and the changes of monocytes and macrophage (Figure 5B) in lung metastasis mice in response to ICI treatment.”

“Among T cells in tumor microenvironment, CD8+ T cells and NKG2D+ T cell were the major cells expressing PD-1 (Figure 5C) and PD-L1 (Figure 5D), respectively.”

Figure 5A shows the alteration of T cells and neutrophils after ICI therapies. Figure 5B represents the non-T cells and non-neutrophils population (CD3-/Ly6G-), including macrophages, monocytes, NK cells, DC, and B cells. The changes of macrophages/monocytes/DC are revealed in the lower right region as CD11b+/Cy6C+ cells. Figure 5C and Figure 5D indicate the changes after ICI treatment in the major PD-1-expressing cells (CD8+ T cells) and PD-L1 expressing cells (NKG2D+ T cells), respectively.

Comment 8: Furthermore, it is not clear why the authors have analyzed the expression of PD1 and PDL-1 on leukocytes in great detail, while no evident expression of PDL-1 is shown for CT26 cell line. Indeed, one can hypothesize that the therapy with anti-PDL-1 in the mouse model can be efficient if this marker is expressed on tumor cells beside leukocytes. Vice versa for the expression of PD1 which should inhibit immune cell response, thus it is relevant to know if this marker is present on lymphocytes and how it is modulated during the therapy.

Response 8: We appreciate the helpful comment to improve our data interpretation. In our flow cytometric analysis for immune cell profiles, the expression of PD-L1 could be seen in various types of immune cells. To simulate the concept of using discriminative expression of PD-L1 in tumor and immune cells in clinical practice, we separately examined the expression of non-leukocytes (CD45+) and non-leukocytes (CD45-) in lung and spleen for a better understanding of post-treatment profiles. Notably, both CT26 tumor cells and stroma cells were included in the non-leukocyte population.

Comment 9: In addition, I would say that the model used consisting of colon cancer cells localized in the lung is not really reminescent of the case report illustrated at the beginning of this paper (a lung adenocarcinoma). This should be pointed out because it is a limitation to transfer the information given to a clinical setting. 

Response 9: The issue for this limitation of our study has been addressed in the Discussion (Line 428) in revised text as follows:

“For validating the finding from a lung adenocarcinoma patient, we applied colon CT26 cancer with lung metastasis model to test the effect of dual blockade in mice. The main reason is that this model has been widely used in immune checkpoint blockade studies, including each blockade for PD-1 and PD-L1, with more comprehensive understanding of dynamic alterations in immune profiles. After this proof-of-concept study, the further investigations by using various types of cancers are warranted.”

Comment 10: Finally, it appears that some relevant references on the topic are missing. Indeed, the works which follow should be cited and discussed.

 1)      Anti-PD1 associated fulminant myocarditis after a single pembrolizumab dose: the role of occult pre-existing autoimmunity. Martinez-Calle N, Rodriguez-Otero P, Villar S, Mejías L, Melero I, Prosper F, Marinello P, Paiva B, Idoate M, San-Miguel J. Haematologica. 2018 Jul;103(7):e318-e321. doi: 10.3324/haematol.2017.185777. Epub 2018 Apr 12. No abstract available

 2)      PD-1 protects against inflammation and myocyte damage in T cell-mediated myocarditis. Tarrio ML, Grabie N, Bu DX, Sharpe AH, Lichtman AH. J Immunol. 2012 May 15;188(10):4876-84. doi: 10.4049/jimmunol.1200389. Epub 2012 Apr 9. PMID:22491251

 3)      Drug-induced myocarditis after nivolumab treatment in a patient with PDL1- negative squamous cell carcinoma of the lung. Semper H, Muehlberg F, Schulz-Menger J, Allewelt M, Grohé C. Lung Cancer. 2016 Sep;99:117-9. doi: 10.1016/j.lungcan.2016.06.025. Epub 2016 Jul 1. PMID:27565924

Response 10: We appreciate the helpful comment to improve the citation in our text. We have added these references in revised version.

1.          Reference 1 is described in the Discussion (Line 475 and ref. 38) as follows:

“Furthermore, myocarditis was also reported in multiple myeloma patient treated with immunomodulatory drug (lenalidomide-dexamethasone) combined with anti-PD-1 inhibitor (pembrolizumab) [38]. Autopsy revealed the increase of infiltrate of macrophages, CD8+ T cells and focal fibrosis in the myocardium after combination therapy.”

2.          Reference 2 is mentioned in the Discussion (Line 442 and ref. 33) as follows:

“Tarrio et al. revealed transfer of ovalbumin-specific CD8+ T cells into cMyc-mOva mice resulted in enhanced immunization accompanied with more myocardial inflammation in recipients receiving PD-1 null T cells [33]. They concluded PD-1 deficient T cells are more efficient killers of target cells and induce more inflammation, as evident with the increase of CD8+ T cells, neutrophils, and macrophages in murine myocardium. In our case, there is no significant induction of CD8+ T cells in the heart, and inflammatory monocyte (Ly6C++), but not macrophages (Ly6C+) are significantly increased on day 11 post-treatment of PD-1 plus PD-L1 blockade in lung metastasis mouse model.”

3.          Reference 3 is listed in the Discussion (Line 449 and ref. 34) as follows:

“Another case report resembled our clinical case mentioned that PD-1 blockade-induced myocarditis was identified in a patient with lung squamous cell carcinoma received simultaneous whole brain radiotherapy [34]. In our case, myocarditis suddenly appeared on day 3 post-treatment of PD-L1 blockade but not in the therapeutic period of PD-1 inhibitor combined with radiotherapy, implying combinatorial use of PD-1 and PD-L1 inhibitors may lead to substantial increase of immunotherapy-induced cardiomyopathy.”

Round 2

Reviewer 2 Report

The Authors have extensively revised the paper according to reviewer's suggestions.